# Does Melatonin Associated with Nanostructured Calcium Phosphate Improve Alveolar Bone Repair?

**DOI:** 10.3390/medicina58121720

**Published:** 2022-11-24

**Authors:** Camila Diuana de Almeida, Suelen Cristina Sartoretto, Adriana Terezinha Neves Novellino Alves, Rodrigo Figueiredo de Brito Resende, Jose de Albuquerque Calasans-Maia, Vittorio Moraschini, Alexandre Rossi, José Mauro Granjeiro, Roberto Sacco, Mônica Diuana Calasans-Maia

**Affiliations:** 1Post-Graduation Program in Dentistry, Dentistry School, Federal Fluminense University, Niteroi 24020-140, Brazil; 2Clinical Research in Dentistry Laboratory, School of Dentistry, Federal Fluminense University, Niteroi 24020-140, Brazil; 3Oral Surgery Department, Dentistry School, Federal Fluminense University, Niteroi 24020-140, Brazil; 4Department of Stomatology, Dentistry School, Federal Fluminense University, Niteroi 24020-140, Brazil; 5Oral Surgery, Dentistry School, Iguaçu University, Nova Iguaçu 26260-045, Brazil; 6Orthodontics Department, Dentistry School, Federal Fluminense University, Niteroi 24020-140, Brazil; 7Department of Condensed Matter, Applied Physics and Nanoscience, Brazilian Center for Research in Physics, Rio de Janeiro 22290-180, Brazil; 8Oral Surgery Department, Division of Dentistry, School of Medical Science, The University of Manchester, Manchester M13 9PL, UK

**Keywords:** melatonin, calcium phosphate, alveolar bone repair, tooth extraction, rats

## Abstract

*Background and objectives*: Calcium phosphates have been widely used as bone substitutes, but their properties are limited to osteoconduction. The association of calcium phosphates with osteoinductive bioactive molecules has been used as a strategy in regenerative medicine. Melatonin has been studied due to its cell protection and antioxidant functions, reducing osteoclastic activity and stimulating newly formed bone. This study aimed to evaluate the effect of topical application of melatonin associated with nanostructured carbonated hydroxyapatite microspheres in the alveolar bone repair of Wistar rats through histological and histomorphometric analysis. *Materials and Methods:* Thirty female Wistar rats (300 g) were used, divided randomly into three experimental groups (*n* = 10), G1: nanostructured carbonated hydroxyapatite microspheres associated with melatonin gel (CHA-M); G2: nanostructured carbonated hydroxyapatite (CHA); G3: blood clot (without alveolar filling). The animals were euthanized after 7 and 42 days of the postoperative period and processed for histological and histomorphometric evaluation. Kruskal–Wallis and Dunn’s post-test were applied to investigate statistical differences between the groups at the same time point for new bone and connective tissue variables. Mann–Whitney was used to assess statistical differences between different time points and in the biomaterial variable. *Results:* Results showed a greater volume of residual biomaterial in the CHA-M than the CHA group (*p* = 0.007), and there were no significant differences in terms of newly formed bone and connective tissue between CHA and CHA-M after 42 days. *Conclusions:* This study concluded that both biomaterials improved alveolar bone repair from 7 to 42 days after surgery, and the association of CHA with melatonin gel reduced the biomaterial’s biodegradation at the implanted site but did not improve the alveolar bone repair.

## 1. Introduction

Melatonin is an endogenous neurohormone that is produced and secreted rhythmically by the pineal gland, controlled by the hypothalamus’ suprachiasmatic nucleus and by the circadian rhythm of light and darkness, affecting sleep [1,2]. The magnitude of circadian rhythm is influenced by age, gender, diseases and hormonal conditions [3]. The hormone’s biosynthesis is initiated through the uptake of the essential amino acid tryptophan by parenchymal cells of the pineal gland. After that, tryptophan is converted into a serotonin neurotransmitter, whose concentration increases during the day and sharply decreases at the beginning of the night, when serotonin is converted into melatonin through the action of the hydroxyindole-O-methyltransferase (HIOMT) enzyme over N-acetyl serotonin (NAS) [4,5,6]. Due to its lipophilic properties, melatonin can cross cellular membranes and bind to cytosolic proteins, such as kinase C, calmodulin and calcireticulin [7]. Several researchers have indicated that melatonin might be dissimilarly distributed at the sub-cellular level. Particularly, cell nuclei and mitochondria appear to hold higher melatonin concentrations than other cellular compartments. Studies have shown that extracellular melatonin equilibrates within seconds with the cytoplasm, confirming that melatonin rapidly crosses biological membranes [5,6,7].

Melatonin has a positive influence in the process of bone remodeling [8] and also carries out diverse neuroendocrine and physiologic functions, for example the control of sleep and circadian rhythm, the regulation of blood pressure, the inhibition of tumoral growth, and immune function. The decline in its production with age is associated with a high incidence of osteoporosis and bone fractures, showing the relationship between melatonin and bone metabolism [9,10], which has been described in previous in vivo studies [11,12,13,14] and occurs in both a direct and indirect manner [15]. First, this hormone directly affects osteoblast differentiation, increasing its cellular proliferation and thus stimulating mineral formation in this cell [15]. Furthermore, there is a rise in expression of bone differentiation markers—such as alkaline phosphatase, osteocalcin and bone sialoprotein—and of osteoblastic genes, such as recombinant human bone morphogenetic protein-2 (rhBMP-2) and recombinant human bone morphogenetic protein-6 (rhBMP-6) [16,17]. Additionally, melatonin also influences osteoclasts directly through the inhibition of osteoclastogenesis [18,19]. This process is completed by the suppression of the activity of the nuclear factor kappa-B ligand (RANKL) receptor activator, through downregulation of formation and osteoblastic activation mediated by RANKL [20,21]. On the other hand, this hormone regulates bone formation indirectly through interaction with systemic hormones, such as parathyroid hormone, calcitonin, and estrogen, extending its effect [3]. Moreover, its antioxidant role captures free radicals released by osteoclasts during the bone resorption process. As a result, melatonin protects bone cells from oxidative attacks carried out by free radicals, thus diminishing the tissue destruction [8,17]. 

The high demand for dental implants in oral rehabilitation, due to various etiologies, induces an extensive search for new biomaterial bone substitutes and new strategies for bone regeneration. Calcium phosphates (CPs), such as hydroxyapatite (HA), have been extensively used in dentistry and medicine for the repair of bone tissue damaged by trauma, tumors, infections, and congenital abnormalities. Calcium phosphate (CaP) particulates, cements, and scaffolds have attracted significant interest as carrier vehicles [21,22,23]. CaP systems have variable stoichiometry, functionality and dissolution properties that make them suitable for cellular delivery [24]. Their chemical similarity to bone and biocompatibility, as well as nanoparticle size, morphology, and porosity contribute to their controlled release properties [22,23,25,26]. The osteoconductivity of CPs or their ability to provide the appropriate scaffold or template for bone formation are well known and have been reported in previous studies [1,2,22,23,25].

Alveolar preservation techniques can help reduce dimensional changes in bone; however, they do not prevent bone resorption. A loss in width of 3–4.8 mm and in height of up to 2.64 mm can be expected [27]. A systematic review showed that horizontal bone loss (29–63%, 3.79 mm after 6 months) was more significant than vertical bone loss (11–22%, 1.24 mm after 6 months). A pattern of rapid resorption in the first 3–6 months and subsequent gradual reduction was observed [28]. Thus, studies with bone substitute biomaterials, such as calcium phosphates associated with osteoinductive agents, are important to preserve the alveolar architecture and allow subsequent implant installation and prosthetic rehabilitation.

Given the above, the aim of this study was to evaluate the osteogenic potential of melatonin gel carried by nanostructured carbonated hydroxyapatite microspheres in alveolar bone repair in rats through histologic and histomorphometric analysis.

## 2. Materials and Methods

### 2.1. Ethical Considerations

Animal experiments and breeding were performed under conditions that were in compliance with the NIH Guide for Care and Use of Laboratory Animals; the Brazilian guidelines for care and utilization of animals for scientific and didactic purposes—DBCA; and the CONCEA euthanasia practice guidelines. The experiments were then approved by the Ethics Commission for Animal Use (CEUA/NAL) under protocol number 5527170719 on 11 October 2019. This study report follows Animal Research: Reporting of in vivo Experiments (ARRIVE) [29] and Planning Research and Experimental Procedures on Animals: Recommendations for Excellence (PREPARE) guidelines where appropriate [30]. This study was conducted according to the 3Rs Program (Reduction, Refinement, and Replacement) guidelines [31], whose objective is to reduce the number of animals used during experimentation and minimize animals’ pain and discomfort. All animals were operated on by the same male surgeon (R.F.B.R.) to avoid any kind of bias.

### 2.2. Animal Model

This study used 30 adults female Wistar rats (age: 5–6 months), with a mean weight of 300/350 g, provided by the Laboratory Animals Centre at Fluminense Federal University, Niterói, Rio de Janeiro, Brazil. The animals were kept in plastic cages (*n* = 2). Each mini-isolator was lined with dry wood shavings (pine wood shavings), a completely harmless, nontoxic and inedible material that was replaced daily to provide favorable conditions and secure animal health and welfare, as well to serve as a thermal insulator and reduce heat conduction from the animals’ bodies. Diet was standard, consisting of granulated feed (Nuvilab^®^, Quimtia Laboratory—Colombo-Paraná-Brazil), changed daily to prevent fungal formation and proliferation via food’s prolonged exposure. Water was supplied ad libitum through a glass container with a stainless-steel spout. Room temperature was maintained between 20 and 22 °C, as this was ideal for animals’ growth, with a controlled light cycle (12 h light and 12 h dark) to provide a correct metabolic cycle. A senior veterinarian provided nutritional recommendations and animal care and ensured pre- and post-operative fasting of animals. Animals were randomly assigned to each group using a system of opaque envelopes to minimize the effects of subjective bias when allocating animals to treatment.

### 2.3. Sample Size Calculation

Sample size was calculated using a power analysis based on results from a previous study, which evaluated newly formed bone in the same experimental animal model and period to estimate the effect size [32]. Five animals were required to have a 90% chance of detection, with significance at the 5% level and an increase in the primary outcome measure (newly bone formed) from 28.2 in the control group to 33.8 in the experimental group. In this study, 30 female Wistar rats, with an average weight of 300 grams, were divided randomly into 3 experimental groups and two periods, distributed across 07 and 42 days after surgical procedures (*n* = 5 group/period) [33].

### 2.4. Synthesis and Physical Chemical Characterization of Nanostructured Carbonated Hydroxyapatite

In this study, microspheres of 425 to 600 μm in diameter, constituted of nanostructured carbonated hydroxyapatite at 37 °C and containing alginate, were used. These materials were synthesized in the Biomaterials Laboratory (LABIOMAT) and characterized physically and chemically through X-ray diffraction (DRX), Scanning Electron Microscopy (SEM), and Fourier-transformed infrared vibrational spectroscopy (FTIR) in the Brazilian Center for Research in Physics. All of these results have been published previously [34,35].

### 2.5. Melatonin Gel

Melatonin gel was prepared through a mix of 1.2 mg of lyophilized melatonin powder and 1.0 mL of propylene glycol (carrier agent), as stated in previous studies [9,36] (Bem Viver Prescription Drug Store—Niterói-Rio de Janeiro-Brazil). The placebo gel was composed of methyl cellulose and a biocompatible solvent (propylene glycol). The only difference between groups was the presence of melatonin.

### 2.6. Surgical Procedures

After a 6-hour fasting, all animals were submitted to surgery under general anesthesia through intraperitoneal access—conforming to the Federal Fluminense University’s experimental animal laboratory (LEA) protocol—with injection of 0.6 mL anesthetic solution, which was prepared with 1.0 mL of ketamine 10% (Dopalen^®^—100 mg/mL; CEVA laboratory—Paulínia-São Paulo-Brazil), 0.5 mL of xylazine 2% (Anasedan^®^—20 mg/mL; CEVA laboratory—Paulínia-São Paulo-Brazil) and 8.5 mL of sterile 0.9% saline solution (KabiPac^®^—Fresenius Kabi-Barueri-São Paulo-Brazil). After anesthesia, a syndesmotomy, luxation and extraction of the maxillary right central incisors was performed using a millimetric probe number 5 (Duflex^®^—Juiz de Fora-Minas Gerais-Brazil), as shown in Figure 1A,B. Afterwards, animals were divided into 3 groups: the first group was filled with 0.06 g of biomaterial microspheres that were embedded in 0.6 mL of melatonin gel for 60 minutes to hydrate the material; the second group was filled with 0.06 g of biomaterial microspheres embedded in 0.6 mL of placebo gel; and the third group did not receive any fillings (blood clot) (Figure 1). Finally, a closure using nylon suture 5-0 (Ethicon^®^—Johnson & Johnson-São Paulo-São Paulo-Brazil) was performed, and an antiseptic was given through gauze embedded in chlorhexidine solution to protect the surgical wound and prevent a secondary infection.

### 2.7. Procedures and Postoperative Care

Anti-inflammatory meloxicam—1 mg/kg (Meloxivet^®^—Duprat-Rio de Janeiro-Rio de Janeiro-Brazil)—was administered through subcutaneous access every 24 hours on the surgery day and kept over the following two days.

### 2.8. Obtaining Samples

Following the experimental periods of 7 and 42 days, five animals in each experimental group were euthanized with a lethal dose of anesthetic (anesthetic overdose), using 200 mg/kg of ketamine (Francotar^®^—Virbac-Jurubatuba-São Paulo-São Paulo-Brazil) and 20 mg/kg of xylazine (Sedazine^®^—Fort Dodge-Rio de Janeiro-Rio de Janeiro-Brazil) in the same syringe, administered through intramuscular access. Collection of samples and surrounding tissues was performed only after absence of vital signs.

### 2.9. Samples Processing

Samples from five animals from the experimental period were fixed in formaldehyde solution 4%, decalcified in bone decalcification solution (Allkimia ^®^—Campinas-São Paulo-Brazil), dehydrated through a series of alcoholic solutions of increasing alcoholic concentration (80–100%), cleared with xylene and embedded in paraffin blocks, cut with a thickness of 5 μm, and stained with hematoxylin–eosin (HE) and with Masson’ trichrome (TM). After that, slices were used for histomorphometric analysis.

### 2.10. Histomorphometric Analysis

For histomorphometric analysis, the photomicrographs were captured from HE-stained slides, with 20× magnification, in a bright field light microscope (OLYMPUS^®^ BX43, Tokyo, Japan) coupled in a high-resolution digital camera (OLYMPUS^®^ SC100, Tokyo, Japan) in CELLSENS^®^ 1.9 software (Digital Image, Olympus, Tokyo, Japan). In each histological section, 6 fields, with no overlapping areas, were captured by scanning corresponding to the interest area into the dental socket.

The photomicrographs were transferred to ImagePro-Plus^®^ 6.0 imaging software (Media Cybernetics, Silver Spring, MD, USA), and a grid of 200 points was superimposed on histological images. Each point of the grid was classified according to the corresponding variable: newly formed bone, biomaterial, and connective tissue, thus allowing the determination of the density of each structure. The number of points for each variable was transformed into a percentage. The average of the fields of each animal was obtained for each variable that was used in the statistical analysis.

### 2.11. Statistical Analysis

After applying the Shapiro–Wilk normality test, the data were considered not normal. A quantitative description for biomaterial, newly formed bone, and connective tissue variables was performed through a non-parametric description with median and interquartile range. Kruskal–Wallis and Dunn’s post-test were applied to investigate statistical differences between the groups at the same time point for new bone and connective tissue variables. Mann–Whitney was used to assess statistical differences between different time points in the biomaterial variable. The variability of the measurements was evaluated with a significance level of 5%. Analysis was performed using Graph Pad PRISM 8.3 software (Inc. La Jolla, San Diego, CA, USA).

## 3. Results

### 3.1. Histological Evaluation

After 07 days, in the clot group (Figure 2A,B), the alveolar socket area was filled by connective tissue containing blood vessels and extravasated red blood cells. The delicate trabeculae of neoformed bone at the defect’s periphery was paved by osteoblasts surrounded by resident mature bone. After 42 days (Figure 2C,D), the central area of the defect presented a small amount of connective tissue surrounded by mature trabeculae of neoformed bone with important osteoblastic activity.

After 07 days, in the group of CHA (Figure 3A,B), the tooth socket area was filled with biomaterial and connective tissue with an important presence of red blood cells. An osteoid matrix was noted in the periphery. After 42 days (Figure 3C,D), the defect area was filled by connective tissue in the central portion containing biomaterial; a large area of the defect was filled by mature trabeculae of neoformed bone with some foci of biomaterial in between.

After 07 days, the CHA-M group (Figure 4A,B) showed the alveolar socket filled with biomaterial and connective tissue with an important presence of red blood cells. After 42 days (Figure 4C,D), the defect area was filled with trabeculae of neoformed bone, sometimes containing biomaterial, interspersed with fibrous connective tissue.

### 3.2. Histomorphometric Results

#### 3.2.1. Newly Formed Bone

Figure 5 presents histomorphometric results of newly formed bone from different experimental groups that were studied 07 and 42 days after implantation. At 07 days, the blood clot group showed more newly formed bone (median 19.33; IQR 16.19–32.32) in relation to the other groups, CHA and CHA-M (median 0.0) (*p* = 0.004). At 42 days, a time-dependent relationship of the increase in newly formed bone in all experimental periods was seen (*p* = 0.007). In this period, the blood clot group (median 61.15; IQR 59.92–69.26) showed better newly formed bone compared to the CHA and CHA-M groups (median 46.54; IQR 31.99–51.02; and 47.37; IQR 36.66–53.68, respectively) (*p* ≤ 0.04).

#### 3.2.2. Biomaterial

After 07 days, no statistical differences were observed between CHA (median 23.05; IQR 16.29–29.81) and CHA-M (median 18.16; IQR 14.71–20.96). However, after 42 days, the CHA-M group (median 4.73; IQR 2.92–6.91) presented greater volume of remaining material compared to CHA (median 1.63; IQR 1.4–1.91) (*p* = 0.007). In both groups, there was a time-dependent decrease in the volume of remaining biomaterial related to a previous period (*p* = 0.007) (Figure 6).

#### 3.2.3. Connective Tissue

A decline in connective tissue’s volume was observed in the clot and CHA-M groups 42 days after surgery, compared to the previous experimental group (*p* = 0.007). In experimental periods, there were no observed differences between groups (Figure 7).

## 4. Discussion

Over the past years, research has assessed a wide variety of calcium phosphates to be used as carriers for drug delivery as antimicrobials [22,23,25], growth factors [34], and bone morphogenic proteins (BMPs) [26] to enhance bone regeneration. Among them, nanostructured carbonated hydroxyapatite has shown better biodegradability, biocompatibility and osteoconduction [32,35,37]. Despite melatonin’s many functions, the most interesting ones for the purpose of bone tissue engineering include: regulation of bone metabolism through the modulation of calcium metabolism; osteoblast differentiation increase; osteoclastogenesis inhibition; interaction with systemic hormones; and antioxidant role [17,18,19,20,38]. The aim of this study was to evaluate the effect of melatonin’s topical application, associated with nanostructured carbonated hydroxyapatite microspheres, in the alveolar bone repair of Wistar rats through histological and histomorphometric analysis. The hypothesis of this study was that the association of this biomaterial with melatonin gel would improve the alveolar bone repair after tooth extraction when compared to the control group. The experimental animal model of this study was adult female Wistar rats, which was different from previous studies that used other experimental animals, such as dogs, rabbits, mice, and Fischer and Sprague Dawley rats [5,15,16,38,39,40,41]. In this study, we only used one sex (female) to avoid bias in the results; this may constitute a limitation of this study since the results cannot be extrapolated to males. All animals were operated on after 5–6 months of age as, at this age, they are considered skeletally mature (adults). Differently from this study, most of the studies previously performed on rats were conducted on tibia or calvaria; for the first time in the literature, alveolar bone repair after tooth extraction in rats was evaluated [13,15,39,40,41,42,43]. Many studies previously performed on tibia investigated the positive effect on new bone formation around implants [4,13,15,16], while others—more like this study—evaluated melatonin’s influence on bone repair in calvaria and tibia [41,44,45]. The same biomaterial reported by previous authors was used in this study [34,35]. Even though many previous studies worked with melatonin, the way it was used in this study differed in terms of concentration and route of administration. Some studies used the same melatonin powder concentration as used here [8,9,42,43], whilst another study used 3 mg of this hormone [5,6,14,15,16,41]. The obtained results showed that the CHA and CHA-M groups improved alveolar bone repair from 7 to 42 days, but there were no statistically significant differences between them in newly formed bone 42 days after grafting. Moreover, over the same experimental period, the melatonin group presented a greater amount of residual biomaterial compared to the control group. This result may suggest that the association of calcium phosphate with melatonin gel delayed the biomaterial absorption process and its maintenance in the sockets may have impaired bone neoformation. In addition, since this research is an unpublished study in the literature, the relationship between the experimental animal model used and the concentration and the administration route of the gel used may influence these results. Although several studies have reported that melatonin acts on bone formation [3,10,12,46,47,48,49,50,51,52], this study did not show results demonstrating its effect on bone regeneration. Systematic review assessed the effectiveness of melatonin’s topical applications in implant dentistry, revealing its direct action on osteoblasts, which induced a higher rate of preosteoblast to osteoblast maturation in terms of quantity and velocity, with a higher rate of osseous matrix production and its corresponding mineralization. Therefore, there was a rise in new bone formation and in bone-to-implant contact (BIC) values around dental implants, leading to a more stable bone environment around them [7]. Moreover, a recent systematic review evaluated the effect of melatonin in periodontal disease through its antioxidant and anti-inflammatory activities, acting directly on osteoclasts. Therefore, there was a neutralization and detoxification of various free radicals produced during osteoclastogenesis, which plays a crucial role in regulating the bone resorption process, thus decreasing periodontitis [53]. Moreover, some investigators hypothesized that the amount of melatonin in saliva and in gingival crevicular fluid seems to be lower in patients that suffer from periodontal disease when compared to the control group, indicating that this hormone may play a protective role in the periodontal tissue [54,55,56,57].

A previous study using melatonin topically observed positive effects on angiogenesis in rabbit tibiae [43]. When applied topically around dental implants associated with growth hormone, it showed a positive effect on bone repair in dogs [42]. Osteoblastic proliferation in rat tibiae [41] and dog dental sockets [39] was also observed after topical application of melatonin. However, studies using melatonin around dental implants did not observe positive effects on bone implant contact after 12 weeks [58] and after 2 weeks [59]. It is possible that the dose of melatonin used in this study may not have favored positive results in bone neoformation when compared to other previous studies. In this study, only an histomorphometric analysis of the middle third of the rat dental socket was conducted, so the hypothesis that melatonin has not remained in the implanted site is not true, mainly due to the anatomy of the rat dental socket, which is favorable to the maintenance of the filling material.

In this study, the authors observed that the association of melatonin gel with calcium phosphate microspheres delayed the degradation of the material. This can be explained by the fact that melatonin has a local anti-inflammatory effect and calcium phosphate degrades faster in a more acidic environment, so the presence of melatonin may have kept the pH neutral, which does not favor degradation. Even if the amount of newly formed bone is the same in the clot group as in the grafted sockets, we know that the presence of the biomaterial preserves the alveolar architecture and prevents the atrophy of the alveolar bone crest, and maintaining adequate contours and dimensions of bone to facilitate the placement of implants in prosthetic driving positions has good esthetic results [60].

## 5. Conclusions

Both biomaterials improved alveolar bone repair from 7 to 42 days. The association of melatonin gel did not improve the alveolar bone repair and did not interfere with biocompatibility and osteoconduction but delayed the biodegradation of carbonated hydroxyapatite microspheres at the implanted site. Future studies should be carried out to explore the possible clinical uses and benefits of this hormone in the treatment of compromised sites and patients.

## Figures and Tables

**Figure 1 medicina-58-01720-f001:**
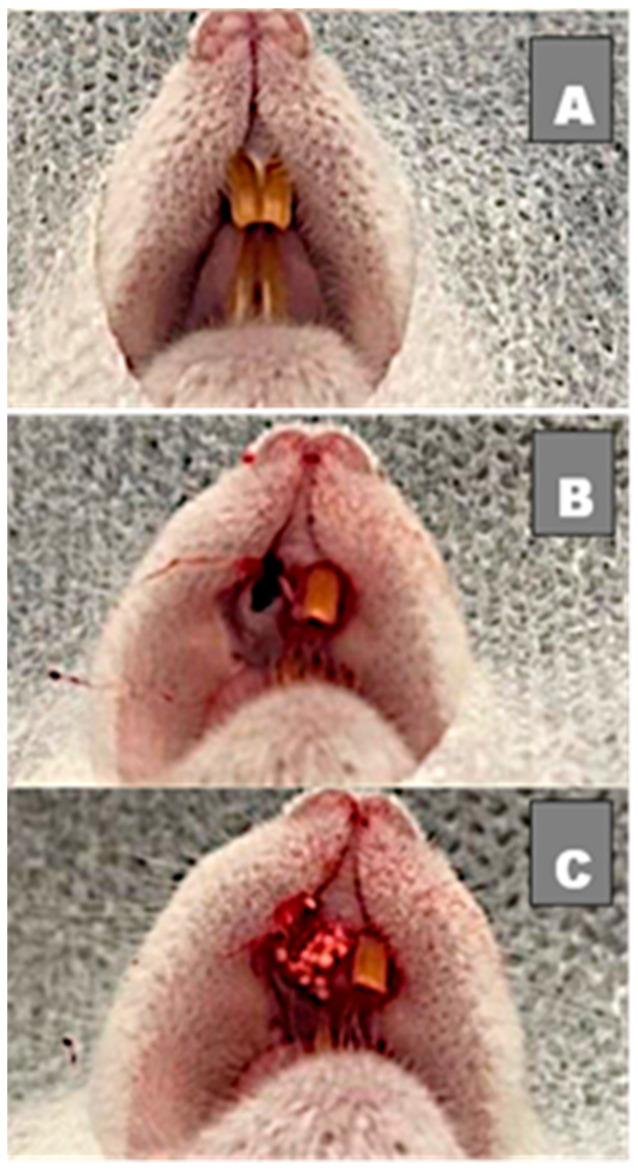
(**A**) Clinical image before surgery with 2 upper incisors; (**B**) Right upper incisor extracted with alveolar bone preserved; (**C**) Socket filled with biomaterial.

**Figure 2 medicina-58-01720-f002:**
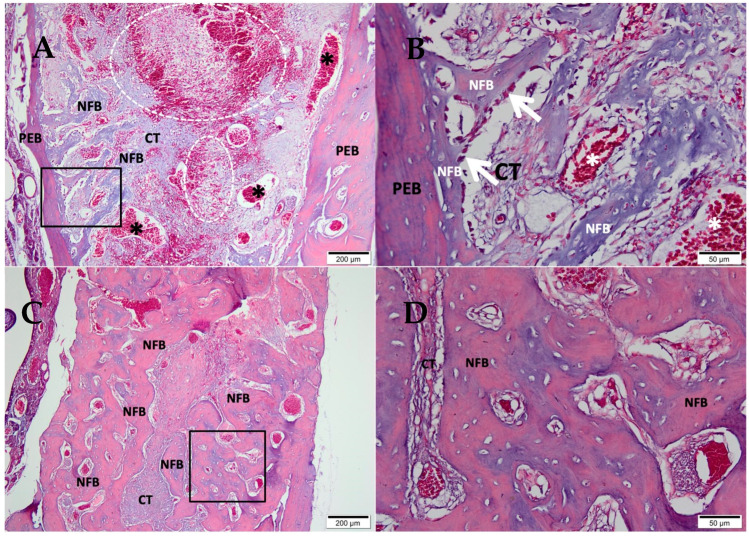
Photomicrographs of the clot group 7 and 42 days after tooth extraction. (**A**,**B**) 07 days; (**C**,**D**) 42 days; (**A**,**C**) 10× magnification of (Bar: 200 μm); (**B**,**D**) 40× magnification of (Bar: 50µm); Stain: Masson’ trichrome; Biomaterial (BM); Connective Tissue (CT); Newly Formed Bone (NFB), Pre-existing Bone (PEB), (*) blood vessels, and (white dotted area) extravasated red blood cells.

**Figure 3 medicina-58-01720-f003:**
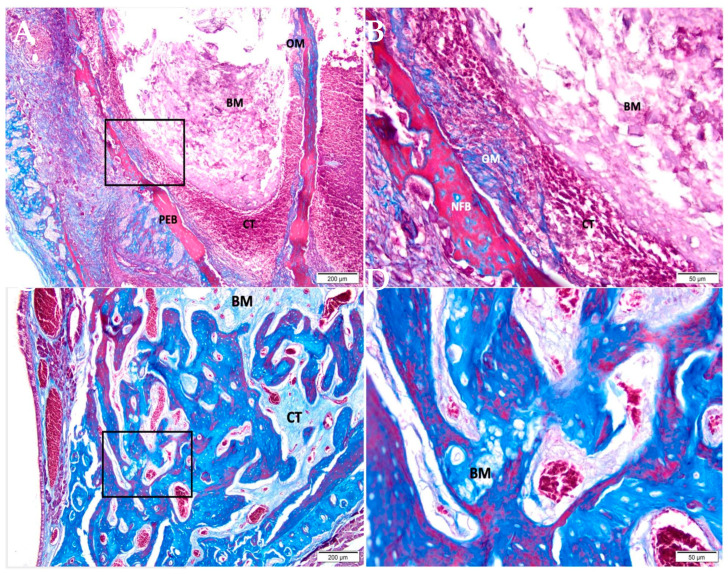
Photomicrographs of the CHA group 7 and 42 days after alveolar grafting. (**A**,**B**) 07 days; (**C**,**D**) 42 days; (**A**,**C**) 10× magnification (Bar: 200 μm); (**B**,**D**) 40× magnification (Bar: 50 μm); Stain: Masson’ trichrome. Biomaterial (BM); Connective Tissue (CT); Pre-Existing Bone (PEB); Newly Formed Bone (NFB).

**Figure 4 medicina-58-01720-f004:**
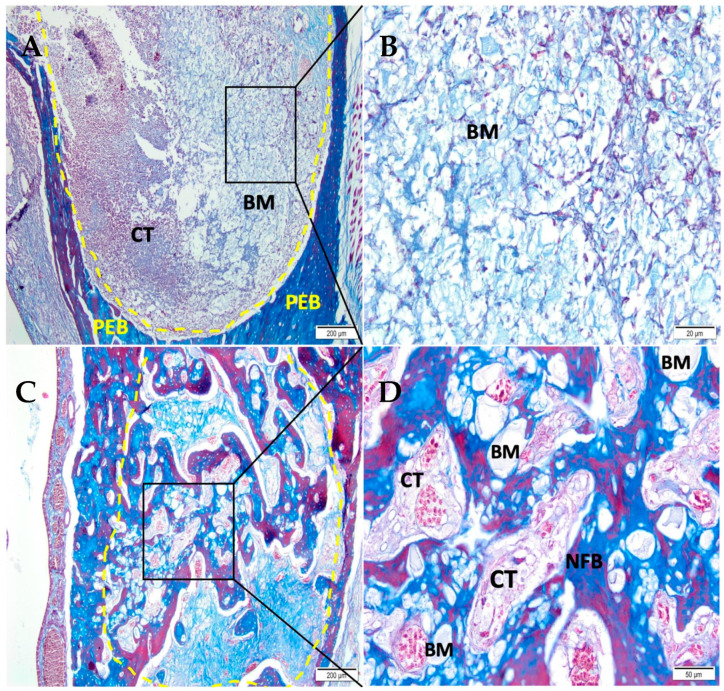
Photomicrographs of the CHA-M group 7 and 42 days after alveolar grafting. (**A**,**B**) 07 days; (**C**,**D**) 42 days; (**A**,**C**) 10× magnification of (Bar: 200 μm); (**B**,**D**) 40× magnification (Bar: 50 µm); Stain: Masson’ trichrome. Biomaterial (BM); Connective Tissue (CT); Newly Formed Bone (NFB), and Pre-Existing Bone (PEB).

**Figure 5 medicina-58-01720-f005:**
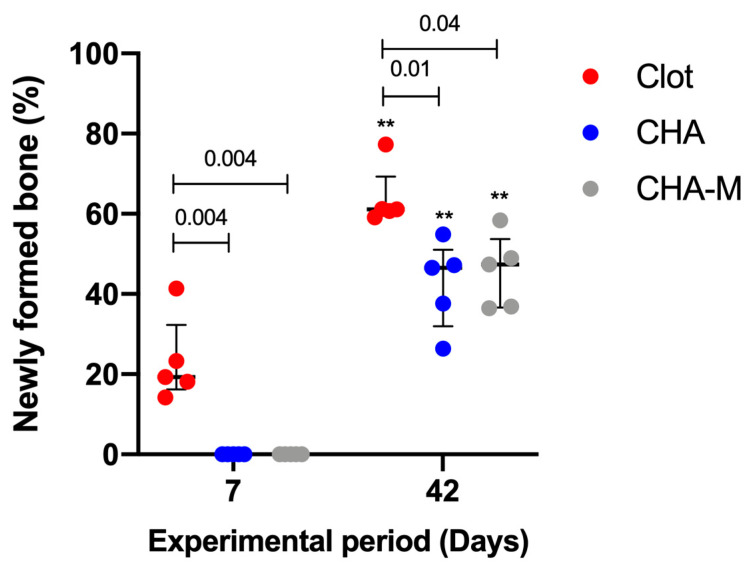
Percentage of newly formed bone of all groups after 7 and 42 days after grafting. Results are presented as a median and IQR (*n* = 5). Kruskal–Wallis and Dunn’s post-test were applied to investigate differences between the groups at the same time point. Mann–Whitney test was used to evaluate differences between the same experimental group in different time points. The (**, *p* = 0.007) represents statistical difference between the same group in different experimental periods. The horizontal bar represents statistical differences between groups over the same period.

**Figure 6 medicina-58-01720-f006:**
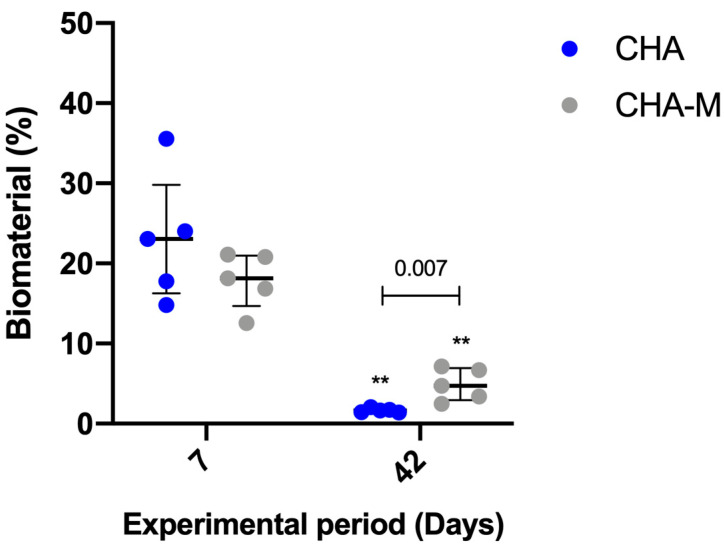
Percentage of biomaterial of CHA and CHA m groups after 7 and 42 days after grafting. Results are presented as a median and IQR (*n* = 5). Kruskal–Wallis and Dunn’s post-test were applied to investigate differences between the groups at the same time point. Mann–Whitney test was used to evaluate differences between groups and time points. The ** *p* = 0.007 represents statistical difference between the same group in different experimental periods. The horizontal bar represents statistical differences between groups over the same period.

**Figure 7 medicina-58-01720-f007:**
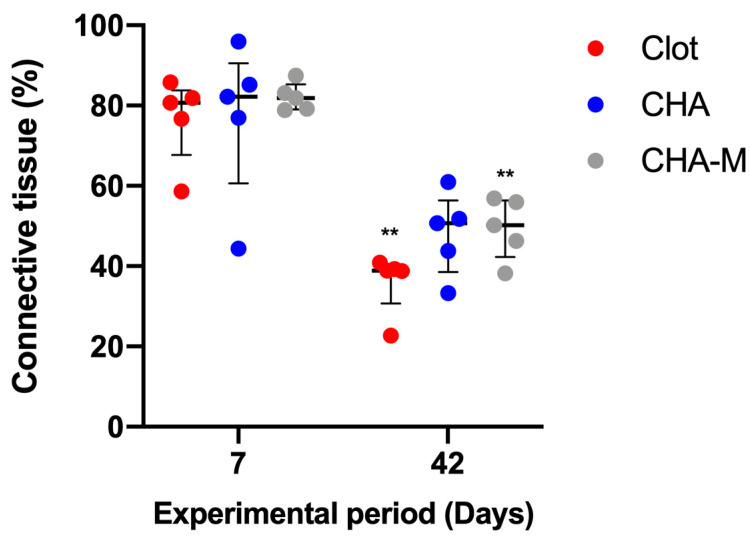
Percentage of connective tissue of all groups after 7 and 42 days after grafting. Results are presented as a median and IQR (*n* = 5). Kruskal–Wallis and Dunn’s post-test were applied to investigate differences between the groups at the same time point. Mann–Whitney test was used to evaluate differences between the same experimental group in different time points. The **, *p* = 0.007 represents statistical difference between the same group in different experimental periods.

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
