# Peer review of "Does Melatonin Associated with Nanostructured Calcium Phosphate Improve Alveolar Bone Repair?"

_medicina, 2022, doi:10.3390/medicina58121720_

Round 1
Reviewer 1 Report
This is an interesting study looking at the effect of melatonin mixed with nanocHA in a model of tooth extraction. The results find that the melatonin did not improve bone formation compared to cHA alone and both showed less bone formation than empty socket. It is really important that we publish negative results and I think these results would be of interest to the scientific community however I feel there are a number of points which need to be address, both major and minor:
1. I think the authors should highlight the lack of effectiveness of melatonin in this study and so the last sentence of the abstract and conclusion at end of the paper should be more clearly and more strongly worded to this effect. Negative results are fine and need to be published so I don’t think it needs to be “spun”.
2. The rationale for choice of the model needs to be clearer in the introduction- especially in light of the fact that so many similar studies with melatonin have been performed. So why this model? What clinical application is it modelling? It isn’t a critical sized defect and the presence of the biomaterial delayed bone formation so how is this relevant to what would happen clinically?
3. Why only female rats used? Need to include the rationale for this in the discussion and postulate if similar results would be expected in male animals
4. I appreciate the thoroughness of the methodological detail to allow repetition and to comply wth ARRIVE guidelines. Recently, it has been highlighted that the sex of the handler during experiments may affect rodents and so in the interests of scientific reproducibility the authors could also consider reporting sex of handler
5. Why was the same carrier agent (propylene glycol) not used for the placebo gel? It seems to be totally different in composition in the cHA group only compared to McHA when they should only differ in terms of the presence of melatonin
6. Pg3 L139-144 it isn’t clear what was done to each biomaterial and they seem different e.g. one hydrated, the other not, 6g of one embedded, 7g of the other? And where was the placebo gel used? And what was in group 3? It is described as a blood clot but was the area flushed before allowing to clot to make sure all the debris was removed following surgery?-
7. Did the implant stay in place for the duration of the experiment in all animals? It seems quite open from the images in figure 1.
8. What does “sent out” for histomorphometric analysis mean? Where to? L169
9. More detail on how quantification of histo was done it just says this “allowed us” to determine but how was the grid applied etc L177
10. Is there a typo? Vacancy analysis should be analysis of variance? L182- Indeed, the whole statistical section could be clearer. When was Kruskal wallis used? Did log transformation make the quantitative data normal? Also check for spelling- in some places in manuscript written as turkey instead of tukey
11. Some figure labels are not in English
12. Don’t need the table and the graphs as they show the same thing
13. Fig 4 C and D are the same images as Figure 5 C and D but are meant should be from different groups
14. More histological images could be included with specific features of interest and some from clot group
15. Discussion needs expanded to develop theories as to why no effect was seen in this model- differences between the studies are adescribed but they don’t state (a) if all the other studies found a positive effect of the melatonin and (b) what the specific reasons could be for lack of effect in this study could be i.e. different strain of rat and different delivery system is not enough. Did the melatonin not stay in place?
Author Response
To: Editor-in-Chief of Medicina: special issue “Advances in Oral Surgery and Implant Dentistry”
Manuscript ID: medicina-1977175
Title: Does melatonin associated with nanostructured calcium phosphate
improve alveolar bone repair?
Thank you for the attention you have given to our manuscript originally entitled “Does melatonin associated with nanostructured calcium phosphate improve alveolar bone repair?” Indeed, we appreciated the comments and criticism of the reviewer and the opportunity you have given us to submit a new revision of our reworked manuscript.
The authors would like to acknowledge the effective and unbiased review of the manuscript, believing that introduction of the suggested alterations produced a manuscript with better editorial and scientific quality.
Please find below the point-by-point the answers to reviewers’ comments:
REVIEW #1
This is an interesting study looking at the effect of melatonin mixed with nanocHA in a model of tooth extraction. The results find that the melatonin did not improve bone formation compared to cHA alone and both showed less bone formation than empty socket. It is really important that we publish negative results and I think these results would be of interest to the scientific community however I feel there are a number of points which need to be address, both major and minor:
- I think the authors should highlight the lack of effectiveness of melatonin in this study and so the last sentence of the abstract and conclusion at end of the paper should be more clearly and more strongly worded to this effect. Negative results are fine and need to be published so I don’t think it needs to be “spun”.
Answer: The authors have included a new sentence at the end of the abstract section and a conclusion that the association with melatonin did not improve alveolar bone repair. This sentence was included, and it is in red.
- The rationale for choice of the model needs to be clearer in the introduction- especially in light of the fact that so many similar studies with melatonin have been performed. So why this model? What clinical application is it modelling? It isn’t a critical sized defect and the presence of the biomaterial delayed bone formation so how is this relevant to what would happen clinically?
Answer: The dental socket model was chosen because there is no published study with alveolar filling with bone substitutes associated with melatonin to evaluate bone repair. The authors included one paragraph with two new references in the Introduction section about the clinical application of this model. The relevance of this study is to show the possibility of the association of two materials for alveolar bone maintenance. The presence of biomaterial did not delay the formation of new bone. Still, in the histomorphometric quantification, due to the presence of biomaterial in the dental socket, the amount of bone is reduced, limited only to the spaces between the spheres of the bone substitute biomaterial.
- Why only female rats used? Need to include the rationale for this in the discussion and postulate if similar results would be expected in male animals.
Answer: In this study, we used only one sex (female) to avoid bias in the results; this may be a study limitation since the results cannot be extrapolated to males. All animals were operated on after 5-6 months of age and, at this age, are already considered skeletally mature (adults). This information was included in the Discussion section.
- I appreciate the thoroughness of the methodological detail to allow repetition and to comply wth ARRIVE guidelines. Recently, it has been highlighted that the sex of the handler during experiments may affect rodents and so in the interests of scientific reproducibility the authors could also consider reporting sex of handler.
Answer: The sex of the handler during the experiments was included in the manuscript (material and methods section).
- Why was the same carrier agent (propylene glycol) not used for the placebo gel? It seems to be totally different in composition in the cHA group only compared to McHA when they should only differ in terms of the presence of melatonin.
Answer: The placebo gel was composed of methylcellulose and a biocompatible solvent (propylene glycol)—the only difference is the presence of melatonin. The author included this information to be more precise for the readers.
- Pg3 L139-144 it isn’t clear what was done to each biomaterial and they seem different e.g. one hydrated, the other not, 6g of one embedded, 7g of the other? And where was the placebo gel used? And what was in group 3? It is described as a blood clot but was the area flushed before allowing to clot to make sure all the debris was removed following surgery?
Answer: The authors thank the reviewer for the correction, there was an editing error in the text and the correct text has been included in red color for easier viewing.
- Did the implant stay in place for the duration of the experiment in all animals? It seems quite open from the images in figure 1.
Answer: Although it seems that the socket was partially open, the area investigated was the middle third of the socket. Due to the anatomy of the rat dental socket, this region allowed the biomaterial to be maintained inside.
- What does “sent out” for histomorphometric analysis mean? Where to? L169
Answer: The authors mean: After that, slices were used for histomorphometric analysis. The “sent out” was substituted by “used”.
- More detail on how quantification of histo was done it just says this “allowed us” to determine but how was the grid applied etc L177
Answer: The authors included this sentence to clarify how the quantification of the points in the histomorphometry was performed: “Each point superimposed on the bone, connective tissue, and residual biomaterial was counted, and at the end of the count, a percentage value was quantified where 200 points are 100%”.
- Is there a typo? Vacancy analysis should be analysis of variance? L182- Indeed, the whole statistical section could be clearer. When was Kruskal wallis used? Did log transformation make the quantitative data normal? Also check for spelling- in some places in manuscript written as turkey instead of tukey.
Answer: Yes, vacancy analysis is an analysis of variance. This term was corrected in the text. Also, the tests used to assess the differences between groups and experimental periods were elucidated. After the data did not pass the normality test (Shapiro-Wilk), the values were transformed into the logarithm of Y. This statistically recognized treatment allows the application of parametric tests. In this case, analysis of variance (ANOVA) and Tukey's post-test were applied to investigate the statistical differences between the different groups and experimental periods for new bone and connective tissue variables. Student's t-test was used to assess statistical differences between other groups and experimental periods in the biomaterial variable. This information was included in the text.
- Some figure labels are not in English.
Answer: The authors substituted the Portuguese labels for English Labels.
- Don’t need the table and the graphs as they show the same thing.
Answer: The authors have removed Table 2 and kept the graphics.
- Fig 4 C and D are the same images as Figure 5 C and D but are meant should be from different groups.
Answer: The authors thank the reviewer for this observation and confirm that there was an error in editing the figure. A new, correct figure has been included for each group.
- More histological images could be included with specific features of interest and some from clot group.
Answer: The authors have included new figure from the clot group (Figure 6).
- Discussion needs expanded to develop theories as to why no effect was seen in this model- differences between the studies are described but they don’t state (a) if all the other studies found a positive effect of the melatonin and (b) what the specific reasons could be for lack of effect in this study could be i.e. different strain of rat and different delivery system is not enough. Did the melatonin not stay in place?
Answer: The authors include a paragraph in the discussion with positive and negative results from previous studies in different animal models. They also include possible justifications for the results obtained in this study.

Reviewer 2 Report
In this paper, the osteogenic potential of the association of melatonin gel and nanostructured carbonated hydroxyapatite microspheres in the alveolar bone repair of rats was studied. This topic is interesting and the work has a very clear structure. But some details still need a lot of work. The introduction section lacks a summary of previous studies to show the novelty of this research. The figures in the results section need to be interpreted further. The discussion section mainly lists the results of previous studies while lacking further exploration and possible reasons for the results.
1. Please summarize the association of melatonin and calcium phosphate/nanostructured biomaterials in previous research in the introduction section.
2. In line 44, please further interpret the effect of the binding of melatonin and cytosolic proteins to improve the coherence of context.
3. The introduction of calcium phosphate is too brief. Please give more information on the osteoconduction of calcium phosphate.
4. Table 2 can be omitted or submitted as an attachment, as the results are already shown in the figures above.
5. Please give an interpretation of Figures 4&5 in the results section.
6. There is no need to repeat the experimental method in the discussion section.
7. The concentrations and routes of administration in previous studies in the discussion section can be shortened as they have little to do with this study.
8. Please discuss the possible reasons that the association of calcium phosphate with melatonin gel delayed the biomaterial absorption process in the discussion section.
9. Please give some prospects for future study in the conclusion section.
10. The standardization of English writing needs to be improved.
Author Response
To: Editor-in-Chief of Medicina: special issue “Advances in Oral Surgery and Implant Dentistry”
Manuscript ID: medicina-1977175
Title: Does melatonin associated with nanostructured calcium phosphate
improve alveolar bone repair?
Thank you for the attention you have given to our manuscript originally entitled “Does melatonin associated with nanostructured calcium phosphate improve alveolar bone repair?” Indeed, we appreciated the comments and criticism of the reviewer and the opportunity you have given us to submit a new revision of our reworked manuscript.
The authors would like to acknowledge the effective and unbiased review of the manuscript, believing that introduction of the suggested alterations produced a manuscript with better editorial and scientific quality.
Please find below the point-by-point the answers to reviewers’ comments:
REVIEW #2
In this paper, the osteogenic potential of the association of melatonin gel and nanostructured carbonated hydroxyapatite microspheres in the alveolar bone repair of rats was studied. This topic is interesting and the work has a very clear structure. But some details still need a lot of work. The introduction section lacks a summary of previous studies to show the novelty of this research. The figures in the results section need to be interpreted further. The discussion section mainly lists the results of previous studies while lacking further exploration and possible reasons for the results.
- Please summarize the association of melatonin and calcium phosphate/nanostructured biomaterials in previous research in the introduction section.
Answer: In the literature, only studies using the association of melatonin with calcium phosphate for osteoporosis in aged rats, melatonin associated with hydroxyapatite-coated dental implants, or in vitro studies using melatonin associated with porous hydroxyapatite have been found. No previous studies have used calcium phosphate associated with melatonin gel for rat alveolar socket filling. The authors included a paragraph in the introduction citing previous studies that used melatonin in different animal models.
- In line 44, please further interpret the effect of the binding of melatonin and cytosolic proteins to improve the coherence of context.
Answer: Thanks for the remark a new paragraph has been added.
- The introduction of calcium phosphate is too brief. Please give more information on the osteoconduction of calcium phosphate.
Answer: The authors included more information about calcium phosphate in the Introduction section.
- Table 2 can be omitted or submitted as an attachment, as the results are already shown in the figures above.
Answer: The authors have removed table 2 and kept the graphics.
- Please give an interpretation of Figures 4&5 in the results section.
Answer: The authors have included a new paragraph with a histological evaluation of the three groups.
- There is no need to repeat the experimental method in the discussion section.
Answer: The authors removed the sentences that described the experimental method in the discussion section.
- The concentrations and routes of administration in previous studies in the discussion section can be shortened as they have little to do with this study.
Answer: The authors removed the text the concentrations and routes of administration in previous studies.
The original text: Even though many previous studies worked with melatonin, our use differed in terms of concentration and route of administration. Some of them had the same melatonin powder concentration as we used 8-9; 40-41, whilst another study used 3 mg of this hormone 14. Histing et al. (2012) treated their animals daily with melatonin (50 mg/kg) for 2 to 5 weeks during the observation period, whereas Koparal et al. (2016) dissolved melatonin (10 mg/kg) in 1 % ethanol just before use and this mixture was injected intraperitoneally. Furthermore, Calvo e Guirrado et al. (2010) used 5 mg of melatonin impregnated in an absorbable sponge followed by a dental implant, while Salomó-Coll et al. (2016) submerged implants in melatonin at 5% in saline solution before its placement. On the other hand, Witt-Enderby et al. (2012) diluted 15 mg/L of melatonin into drinking water, which was provided to mice during hours of darkness.
The revised text: Even though many previous studies worked with melatonin, our use differed in terms of concentration and route of administration. Some of them used the same melatonin powder concentration as we used 8-9; 42-43, whilst another study used 3 mg of this hormone 5, 6,14-16, 41.
- Please discuss the possible reasons that the association of calcium phosphate with melatonin gel delayed the biomaterial absorption process in the discussion section.
Answer: Thanks for the remark. The authors have included a paragraph in the discussion.
- Please give some prospects for future study in the conclusion section.
Answer: The authors included that future studies should be carried out to explore the possible clinical uses and benefits of this hormone in the treatment of compromised sites and patients in the Conclusion section.
- The standardization of English writing needs to be improved.
Answer: The authors forwarded the manuscript for English proofreading.

Round 2
Reviewer 1 Report
Thank you for addressing my comments- a few follow up points:
- The rationale for choice of the model needs to be clearer in the introduction- especially in light of the fact that so many similar studies with melatonin have been performed. So why this model? What clinical application is it modelling? It isn’t a critical sized defect and the presence of the biomaterial delayed bone formation so how is this relevant to what would happen clinically?
Answer: The dental socket model was chosen because there is no published study with alveolar filling with bone substitutes associated with melatonin to evaluate bone repair. The authors included one paragraph with two new references in the Introduction section about the clinical application of this model. The relevance of this study is to show the possibility of the association of two materials for alveolar bone maintenance. The presence of biomaterial did not delay the formation of new bone. Still, in the histomorphometric quantification, due to the presence of biomaterial in the dental socket, the amount of bone is reduced, limited only to the spaces between the spheres of the bone substitute biomaterial.
When I said that the bone healing was delayed I was specifically referring to the fact that there was more bone in the empty defect group therefore the defect was obviously not critical sized. If the presence of the biomaterial reduces the amount of bone then why not just leave these defects to heal? How does the model specifically relate to what would be experienced clinically and the specific clinical application?
- More detail on how quantification of histo was done it just says this “allowed us” to determine but how was the grid applied etc L177
Answer: The authors included this sentence to clarify how the quantification of the points in the histomorphometry was performed: “Each point superimposed on the bone, connective tissue, and residual biomaterial was counted, and at the end of the count, a percentage value was quantified where 200 points are 100%”.
This isn’t at all clear as to how this was performed.
- Is there a typo? Vacancy analysis should be analysis of variance? L182- Indeed, the whole statistical section could be clearer. When was Kruskal wallis used? Did log transformation make the quantitative data normal? Also check for spelling- in some places in manuscript written as turkey instead of tukey.
Answer: Yes, vacancy analysis is an analysis of variance. This term was corrected in the text. Also, the tests used to assess the differences between groups and experimental periods were elucidated. After the data did not pass the normality test (Shapiro-Wilk), the values were transformed into the logarithm of Y. This statistically recognized treatment allows the application of parametric tests. In this case, analysis of variance (ANOVA) and Tukey's post-test were applied to investigate the statistical differences between the different groups and experimental periods for new bone and connective tissue variables. Student's t-test was used to assess statistical differences between other groups and experimental periods in the biomaterial variable. This information was included in the text.
Following logarithmic transformation you need to check for normality again. If it is still not normal then you shouldn’t use a parametric test.
Author Response
Thank you for the attention you have given to our manuscript originally entitled “Does melatonin associated with nanostructured calcium phosphate improve alveolar bone repair?” Indeed, we appreciated the comments and criticism of the reviewer and the opportunity you have given us to resubmit a new revision of our reworked manuscript.
The authors would like to acknowledge the effective and unbiased review of the manuscript, believing that introduction of the suggested alterations produced a manuscript with better editorial and scientific quality.
Please find below the point-by-point the answers to reviewer’ comments:
REVIEW #1
- When I said that the bone healing was delayed, I was specifically referring to the fact that there was more bone in the empty defect group therefore the defect was obviously not critical sized. If the presence of the biomaterial reduces the amount of bone, then why not just leave these defects to heal? How does the model specifically relate to what would be experienced clinically and the specific clinical application?
Answer: In the healing phase after extraction, alveolar bone undergoes additional atrophy because of the natural remodeling process. This begins immediately after extraction and may result in up to 50 % resorption of the alveolar ridge width even in 3 months [1]. Post-extraction alveolar ridge resorption may have an impact on dental implant placement, since sufficient vertical and horizontal volume of alveolar bone should ideally be present at the site of insertion [2]. Bone substitute biomaterial has been used to prevent atrophy of the alveolar bone crest and to maintain adequate contours and dimensions of bone to facilitate the placement of implants in prosthetic driving positions and with good esthetic results [3]. Even if the amount of newly formed bone is the same inside an empty socket as in a grafted socket, the presence of the biomaterial preserves the alveolar architecture vertically and horizontally.
- Schropp L, Wenzel A, Kostopoulos L, Karring T (2003) Bone healing and soft tissue contour changes following single-tooth extraction: a clinical and radiographic 12-month prospective study. Int J Periodontics Restor Dent 23:313–323
- Albrektsson T, Brånemark PI, Hansson HA, Lindström J (1981) Osseointegrated titanium implants: requirements for ensuring a long-lasting, direct bone-to-implant anchorage in man. Acta Orthop 52:155–170.
- Attila Horváth, Nikos Marda, Luis André Mezzomo, Ian G. Needleman, Nikos Donos. (2012) Alveolar ridge preservation. A systematic review. Clin Oral Invest DOI 10.1007/s00784-012-0758-5.
- More detail on how quantification of histo was done it just says this “allowed us” to determine but how was the grid applied etc L177. This isn’t at all clear as to how this was performed.
Answer: The authors have included details of the histomorphometric analysis as follows:
Included text: “For histomorphometric analysis, the photomicrographs were captured from HE stained slides, with 20 X magnification, in a bright field light microscope (OLYMPUS® BX43, Tokyo, Japan) coupled in a high-resolution digital camera (OLYMPUS® SC100, Tokyo, Japan) in CELLSENS® 1.9 software (Digital Image, Olympus, Tokyo, Japan). In each histological section, 6 fields, with no overlapping areas, were captured by scanning corresponding to the interest area into the dental socket.
The photomicrographs were transferred to ImagePro-Plus® 6.0 imaging software (Media Cybernetics, Silver Spring, Maryland, USA), and a grid of 200 points was superimposed on histological images. Each point of the grid was classified according to the corresponding variable: newly formed bone, biomaterial, and connective tissue, thus allowing the determination of the density of each structure. The number of points for each variable was transformed into a percentage. The average of the fields of each animal was obtained for each variable that was used in the statistical analysis.”
- Following logarithmic transformation you need to check for normality again. If it is still not normal then you shouldn’t use a parametric test.
Answer: After the logarithmic transformation, the data didn’t present a normal distribution. We applied the non-parametric tests in the original obtained values. The new statistical analysis was included in the methodology section as follows:
Included text: “After applying the Shapiro-Wilk normality test, the data were considered not normal. A quantitative description for biomaterial, newly formed bone, and connective tissue variables was performed through a non-parametric description with median and interquartile range. Kruskal-Wallis and Dunn’s post-test were applied to investigate statistical differences between the groups at the same time point for new bone and connective tissue variables. Mann-Whitney was used to assess statistical differences between different time points, also in the biomaterial variable. The variability of the measurements was evaluated with a significance level of 5%. Analysis was performed using Graph Pad PRISM 8.3 software (Inc. La Jolla, CA, USA).”
After the new statistical analysis, few changes were observed and did not change the discussion and conclusion of the work. The new results were included and the graphs and description of the histomorphometric results were changed.

Reviewer 2 Report
The author appropriately responded to all questions and the article can be accepted after the following two errors are corrected.
1. In the histological evaluation section, please make the order of description of each group correspond with the order of figures.
2. The expression is repetitive in line 167-174.
Author Response
Thank you for the attention you have given to our manuscript originally entitled “Does melatonin associated with nanostructured calcium phosphate improve alveolar bone repair?” Indeed, we appreciated the comments and criticism of the reviewer and the opportunity you have given us to resubmit a new revision of our reworked manuscript.
The authors would like to acknowledge the effective and unbiased review of the manuscript, believing that introduction of the suggested alterations produced a manuscript with better editorial and scientific quality.
Please find below the point-by-point the answers to reviewer’ comments:
REVIEW #2
- In the histological evaluation section, please make the order of description of each group correspond with the order of figures.
Answer: The authors made the order of histological evaluation according to the order of figures.
- The expression is repetitive in line 167-174.
Answer: The authors have corrected the repetitive text.
